

# Effects of Gas-Wall Partitioning in Teflon Tubing and Instrumentation on Time-Resolved Measurements of Gas-Phase Organic Compounds

Demetrios Pagonis[1,2], Jordan E. Krechmer[1,2,3], Joost de Gouw[1,4], Jose L. Jimenez[1,2], and Paul J. Ziemann[1,2]

[1]Cooperative Institute for Research in Environmental Sciences (CIRES), Boulder, Colorado 80309, United States
[2]Department of Chemistry and Biochemistry, University of Colorado, Boulder, Colorado 80309, United States
[3]Aerodyne Research, Inc., Billerica, Massachusetts 01821, United States
[4]NOAA Earth System Research Laboratory, Boulder, Colorado 80305, United States

Correspondence to: Paul J. Ziemann (paul.ziemann@colorado.edu) or Jose L. Jimenez (jose.jimenez@colorado.edu)

**Abstract.**

Recent studies have demonstrated that organic compounds can partition from the gas phase to the walls in Teflon environmental chambers, and that the process can be modeled as absorptive partitioning. Here these studies were extended to investigate gas-wall partitioning of organic compounds in Teflon tubing and inside a proton transfer reaction-mass spectrometer (PTR-MS) used to monitor compound concentrations. Rapid partitioning of $C_8$–$C_{14}$ 2-ketones and $C_{11}$–$C_{16}$ 1-alkenes was observed for compounds with saturation concentrations ($c^*$) in the range of $3 \times 10^4$ to $1 \times 10^7$ $\mu g$ $m^{-3}$, causing delays in instrument response to step-function changes in the concentration of compounds being measured. These delays vary proportionally with tubing length and diameter and inversely with flow rate and $c^*$. The gas-wall partitioning process that occurs in tubing is similar to what occurs in a gas chromatography column, and the measured delay times (analogous to retention times) were accurately described using a linear chromatography model where the walls were treated as an equivalent absorbing mass that is consistent with values determined for Teflon environmental chambers. The effect of PTR-MS surfaces on delay times was also quantified and incorporated into the model. The model predicts delays of an hour or more for semivolatile compounds measured under commonly employed conditions. These results and the model can enable better quantitative design of sampling systems, in particular when fast response is needed, such as for rapid transients, aircraft, or eddy covariance measurements. They may also allow estimation of $c^*$ values for unidentified organic compounds detected by mass spectrometry, and could be employed to introduce differences in time series of compounds for use with factor analysis methods. Best practices are suggested for sampling organic compounds through Teflon tubing.





## 1 Introduction

Teflon tubing is widely used for sampling organic gases in field and laboratory studies, primarily because it is chemically inert and flexible. These properties also make Teflon the material of choice for environmental "smog" chambers, most of which are constructed using fluorinated ethylene-propylene (FEP) or perfluoroalkoxy (PFA) Teflon film (Hallquist et al.

2009). Although it has been known for decades that Teflon is permeable to small organic compounds (Yi-Yan et al., 1980; Chemours, 2016), only recently have environmental chamber studies shown that it can also absorb large gaseous organic compounds in an equilibrium partitioning process that is rapid (time scale ~10–60 min), reversible, and independent of the age of the chamber (Matsunaga and Ziemann, 2010; Yeh and Ziemann, 2015; Zhang et al., 2015; Krechmer et al., 2016; Ye et al., 2016). The equilibrium reached in this process can be conveniently described using a model that is analogous to gas-

particle partitioning theory (Matsunaga and Ziemann, 2010), in which the chamber walls are treated as an equivalent mass concentration of liquid organic aerosol, $C_w$. Values of $C_w$ reported by Matsunaga and Ziemann (2010), Yeh and Ziemann (2015), and Krechmer et al. (2016) range from 0.3-30 mg m$^{-3}$ for an 8 m$^3$ chamber, a range indicating that a significant fraction of organic products formed from oxidation reactions regularly studied in environmental chambers (and even some less-volatile precursors) will be absorbed into the walls at equilibrium. Using typical values for $C_w$ and the time scale for

reaching gas-wall partitioning equilibrium, one can incorporate the effect into box models to estimate the effect of partitioning on chamber measurements, as has been done in several studies of secondary organic aerosol (SOA) yields (Matsunaga and Ziemann, 2010; Shiraiwa et al., 2013; McVay et al., 2014; Bian et al., 2015; Krechmer et al., 2015; La et al., 2016).

Although gas-wall partitioning of organic compounds in Teflon environmental chambers has now been systematically

investigated in a number of studies, this is not the case for Teflon tubing used for sampling lines. Delays in instrument response have been observed and documented repeatedly, with many references to "sticky" compounds and "memory effects" inside tubing and instrumentation. For example: Teflon O-rings are used inside the proton transfer reaction-mass spectrometer (PTR-MS) because other materials caused significant delays (Warneke et al., 2003); measured eddy covariance frequencies are dampened by sticky compounds (Park et al., 2013); heated instruments have been developed specifically to

reduce delays for semivolatile compounds (Mikoviny et al., 2010); and instrument surfaces are often heated when measuring aerosol components through thermal desorption (Holzinger et al., 2010). Memory effects in the inlet of the PTR-MS have been modeled previously, but the concept was not extended to tubing (Schuhfried et al., 2012).

In the study presented here, we quantified delays observed when a set of organic compounds with a range of volatilities were sampled through Teflon tubing for analysis in a PTR-MS, and then developed a model that applies the principles of gas

chromatography and gas-wall partitioning in Teflon environmental chambers to predict the delays measured for different tubing lengths and diameters, flow rates, and organic functional groups. The results quantify the potential effects of tubing on measurements of organic gases and enable better design of sampling systems, in particular when fast instrument response is needed.



## 2 Experimental section

### 2.1 Measurements of tubing delay

Experiments were conducted by sampling homologous series of either $C_{11}$, $C_{12}$, $C_{14}$–$C_{16}$ 1-alkenes (1-undecene, 1-dodecene, 1-tetradecene, 1-pentadecene, 1-hexadecene) or $C_8$, $C_{10}$, $C_{12}$–$C_{14}$ 2-ketones (2-octanone, 2-decanone, 2-dodecanone, 2-

tridecanone, 2-tetradecanone) from an 8 $m^3$ FEP Teflon environmental chamber into a quadrupole PTR-MS using Teflon tubing. The PTR-MS has been described previously (de Gouw and Warneke, 2007). For each experiment approximately 20 ppb of each of the compounds in a homologous series were added to the chamber by evaporating a known amount from a glass bulb (with heating as necessary) into a 5 L min$^{-1}$ stream of ultra high purity (UHP) $N_2$. The chamber was then mixed with a Teflon-coated fan for 1 min and allowed to sit for 30 min to ensure that gas-wall partitioning equilibrium had been

achieved inside the chamber (Matsunaga and Ziemann, 2010). The combined passivation time of the tubing + PTR-MS for compounds in the chamber was determined by moving the tubing from sampling room air (a clean air source for the conditions of these experiments) to sampling chamber air containing the compounds. This procedure produces a step function in the concentration of compounds sampled, and is identical to frontal analysis, a long-standing technique for characterizing chromatography columns (James and Phillips, 1954; Schay and Szekely, 1954). After the instrument response

reached steady state (meaning that the PTR-MS and tubing were fully equilibrated with the incoming air), either the tubing was moved back to sampling room air, allowing the PTR-MS and tubing to approach a new equilibrium state, or the PTR-MS was investigated separately by detaching the inlet tubing so that it sampled room air directly. The effect of the tubing on the equilibration time was isolated by comparing the response times when the PTR-MS was sampling room air with and without the tubing.

The base case measurements of delays were conducted with 2-ketones sampled through 1.0 m of PFA Teflon tubing (1/4 in. OD, 3/16 in. ID) at a flow rate of 0.36 L min$^{-1}$, with the 1-alkenes evaluated under the same conditions. The effect of tubing length on the delay was evaluated by also using 3.0 m of PFA Teflon tubing with the same OD and ID at a flow rate of 0.36 L min$^{-1}$. The effect of flow rate on the delay was evaluated by increasing the flow rate from 0.36 to 2.7 L min$^{-1}$ by adding a line sampling an additional 2.3 L min$^{-1}$ flow (controlled by a critical orifice) in parallel with the PTR-MS. In this experiment

a 3.0 m length of PFA Teflon tubing was used to achieve a sufficiently large delay. The effect of tubing diameter on the delay was investigated using a 3.0 m length of 1/8 in. OD, 1/16 in. ID PFA Teflon tubing at a flow rate of 0.36 L min$^{-1}$. Flow was laminar in the tubing during all experiments, with calculated Reynolds numbers of 90 and 650 for the low and high flow experiments in the 3/16 in. ID tubing and 260 in the experiment using 1/16 in. ID tubing. The tests were conducted at ambient laboratory temperature (23 °C) and when the instruments had been pumped down and operated for several weeks,

thus representing typical operating conditions.



## 2.2 Chemicals

The following chemicals, purities, and suppliers were used in this study: 1-undecene (97%), 1-dodecene (95%), 1-tetradecene (92%), 1-pentadecene (98%), 2-octanone (98%), 2-decanone (98%), and 2-tridecanone (99%) from Aldrich; 2-dodecanone (98%) and 2-tetradecanone (98%) from ChemSampCo; and 1-hexadecene (99.8%) from Fluka.

## 2.3 Model for transport of an organic compound through Teflon tubing

The model used to describe the effect of tubing on the delay is a linear kinetic chromatography model, where the affinity of a compound for the walls of the Teflon tubing (the stationary phase) is determined by its saturation concentration ($c^*$). This approach seems reasonable, considering the nature of the processes involved, the dependence of gas-wall partitioning on $c^*$ (Matsunaga and Ziemann 2010; Krechmer et al., 2016), and the observation that the extent of partitioning of an organic

compound in a Teflon chamber correlates well with its retention time measured by gas chromatography (Yeh and Ziemann, 2015). Based on the chamber results, we assume that the rate of absorption of a compound into the walls is controlled by gas-phase diffusion to the walls (and thus does not depend on mass accommodation), and treat absorption and desorption as first-order processes. For a numerical solution the tubing is divided into a series of perfectly mixed bins, with compound flowing into and out of each bin and also undergoing gas-wall partitioning, as shown in Fig. 1.

Our assumption of diffusion-limited absorption is consistent with the criteria developed by McMurry and Stolzenburg (1987), who compared time scales for diffusion and uptake at the walls to determine whether mass accommodation affects the uptake kinetics of sticky compounds passing through tubing. They estimate the time scale for diffusion ($\tau_{diff}$) as in Eq. (1):

$$\tau_{diff} = \frac{d_t^{\,2}}{8\,D_g} \tag{1}$$

and the time scale for accommodation into the wall ($\tau_{ac}$) as in Eq. (2):

$$\tau_{ac} = \frac{d_t}{2\,\alpha\,\bar{c}} \tag{2}$$

where $D_g$ is the compound's diffusion coefficient in air, $\alpha$ is its mass accommodation coefficient on the tubing wall, $\bar{c}$ is its mean thermal speed, and $d_t$ is the ID of the tubing. The mass accommodation coefficient has an impact on the rate of uptake at the walls when the time scale for accommodation is comparable to or larger than the time scale for diffusion, as in Eq. (3)

$$\frac{\tau_{ac}}{\tau_{diff}} = \frac{4\,D_g}{\alpha\,\bar{c}\,d_t} \gtrsim 1 \tag{3}$$

Using a diffusion coefficient of 0.067 cm$^2$ s$^{-1}$, the average of values calculated for the compounds studied here (range = 0.055 − 0.088 cm$^2$ s$^{-1}$) using three methods (Tucker and Nelken, 1986), and a tubing ID of 0.47 cm, Eq. (3) indicates that mass accommodation does not affect the rate of uptake to the walls for values of $\alpha > 3 \times 10^{-5}$. Since this threshold is similar to that determined in studies of gas-wall partitioning in Teflon environmental chambers, where it has been shown that the

rate of turbulent mixing within the chamber is the rate-limiting process in establishing partitioning equilibrium for



compounds with mass accommodation coefficients greater than ~$10^{-5}$ (Matsunaga and Ziemann, 2010; Krechmer et al., 2016), our assumption that absorption of compounds into the walls is limited by gas-phase diffusion seems justified.

The first-order rate constant for absorption of compounds into the tubing walls ($k_a$) was calculated using Eq. (4):

$$k_a = \frac{8\,D_g}{d_t^2} \tag{4}$$

which is the inverse of the time scale for diffusion-limited transport to the walls given by Eq. (1). For the 0.47 cm ID tubing used in our experiments, $\tau_a = 0.4$ s. Rate constants for desorption of compounds out of the walls ($k_d$) were calculated using Eq. (5):

$$k_d = \frac{k_a}{K_{gw}} \tag{5}$$

where $K_{gw}$ is the equilibrium constant for gas-wall partitioning inside the tubing. The most and least volatile compounds
measured here had desorption time scales of 0.2 s and 50 s, respectively, in 0.47 cm ID tubing. Values of $K_{gw}$ were calculated using Eq. (6):

$$K_{gw} = \frac{C_w}{c*} \tag{6}$$

which was employed by Matsunaga and Ziemann (2010) in their model for gas-wall partitioning in Teflon environmental chambers. Values of $c*$ were estimated using the SIMPOL.1 group contribution method (Pankow and Asher, 2007), and $C_w$
was obtained by linear fitting of our largest data set (0.47 cm ID tubing sampling at 0.36 L min$^{-1}$) in log-log space using orthogonal distance regression, which gave an optimum value of 4 g m$^{-3}$.

For perfectly mixed flow in the bins, flow is modeled as a first-order process as in Eq. (7):

$$k_f = \frac{Q}{A\,l} \tag{7}$$

where $Q$ is the volumetric flow rate inside the tubing, $A$ is the cross-sectional area of the tubing, and $l$ is the length of a bin in
the model (2 cm). A comparison of the concentration profiles of compounds at the tubing exit simulated assuming perfectly mixed or laminar flow is presented in Fig. S1. The error this assumption causes in transfer time through a 1 m length of tubing is generally smaller than the effect of gas-wall partitioning within the tubing and so does not affect the model results presented here.

Diffusion of compounds absorbed into the walls of the Teflon tubing is fast compared to the time scales
investigated here, and thus it is not explicitly included in our model. This is similar to the assumption often made in chromatography models that diffusion within the stationary phase does not affect mass transfer within the column (Guiochon et al., 2006). Using the formulation developed by Krechmer et al. (2016), we estimate that the depth to which organic compounds effectively partition into the Teflon tubing is 2.2 nm for the $C_w$ value derived here. This value is consistent with our estimates for Teflon chambers, which ranged from 1.5 – 4.5 nm (Krechmer et al., 2016). Diffusion coefficients of larger
organic molecules in Teflon (e.g., toluene and benzene) are $D_t \sim 2 \times 10^{-9}$ cm$^2$ s$^{-1}$ (Tokarev et al., 2006), resulting in an time scale for diffusion in the walls of $\tau_{dw} \sim l_w^2 / D_t \sim 0.1$ ms. This is much smaller than the minimum time scale for gas-phase



diffusion and accommodation of ~400 ms, indicating that this process is too fast to limit partitioning and thus does not need explicit representation in the model.

Using the rate constants defined above, the rates of change in concentration of compounds in the gas phase and wall compartments in bin $i$, $[G_i]$ and $[W_i]$, can be expressed as in Eqs. (8) and (9):

$$\frac{d[G_i]}{dt} = k_f[G_{i-1}] - k_f[G_i] - k_a[G_i] + k_d[W_i] \qquad (8)$$

$$\frac{d[W_i]}{dt} = k_a[G_i] - k_d[W_i] \qquad (9)$$

where both $[G_i]$ and $[W_i]$ are expressed in units of moles per cubic meter of air. These units are consistent with our treatment of gas-wall partitioning as being analogous to gas-particle partitioning, where the concentration of compounds in the condensed phase is represented as moles (or mass) per cubic meter of air (Pankow 1994, Donahue et al., 2006).

The model was solved numerically using the Euler method with at a time step of 1 ms and 50 bins per meter of tubing, using IGOR Pro (Wavemetrics, v7.02). Shortening the time step and/or increasing the number of bins per meter of tubing did not appreciably change the numerical results.

# 3 Results and discussion

## 3.1 Effect of volatility, tubing length, and flow rate on tubing delays

Tubing delays were measured by introducing step function changes in the concentration of organic compounds measured by the PTR-MS, with all compounds of a homologous series being measured simultaneously. We quantify delays in this study as the amount of time required for the PTR-MS signal to achieve 90% of the total change caused by the step function change in sample concentration. As can be seen in Fig. 2A for 2-ketones, the total (PTR-MS + tubing) delay increases with increasing compound carbon number and therefore decreasing compound volatility. This can be explained by noting that the criteria for gas-wall partitioning equilibrium is that the rates of absorption and desorption are equal throughout the system, so for tubing (a similar equation holds for the PTR-MS) this condition is everywhere given by Eq. 10:

$$k_a[G] = k_d[W] \qquad (10)$$

Substituting Eqs. (5) and (6) and rearranging, the equilibrium condition is then given by Eq. 11:

$$[W] = \frac{C_w[G]}{c^*} \qquad (11)$$

Since $C_w$, $[G]$ (the input concentration), and the rate of absorption are essentially the same for all compounds, the time required for $[W]$ to reach the equilibrium value given by Eq. 11 increases with decreasing compound volatility. This reflects the need for the tubing to absorb a larger amount of the less volatile compounds to reach equilibrium, while only a very small amount of the more volatile species needs to be absorbed to meet that condition.

The effects of the PTR-MS and tubing were uncoupled by comparing the equilibration times for the PTR-MS + tubing with the equilibration time of the PTR-MS alone. The differences in equilibration times are significant and easily observed in the time profiles of 2-ketones shown in Fig. 2B. Here we define the tubing delay as the difference in the time it takes for the



signal to drop to 10% of its initial value, with and without the tubing attached to the PTR-MS. Both tubing and instrument delays are substantial.

The tubing delays measured for 2-ketones sampled through two lengths of 0.47 cm ID tubing (1 and 3 m) at a single flow rate (0.36 L min$^{-1}$) and through a single length of 0.47 cm ID tubing (3 m) at two flow rates (0.36 and 2.7 L min$^{-1}$) are shown

in Fig. 3A and Fig. 3B, respectively. The tubing delay increases almost proportionally with tubing length, similar to the effect of column length on retention time in chromatography captured in Eq. (12):

$$t_r = \frac{B\,L}{v_f} \tag{12}$$

where $t_r$ is retention time, $L$ is column length, $v_f$ is flow velocity, and $B$ is a constant that incorporates the partitioning coefficient and volumes of stationary and mobile phases (Skoog et al., 2007). The tubing delay decreases as flow rate is

increased (Fig. 3B) because compounds have less time to partition to the walls, but the observed change is less than inversely proportional to the flow velocity (0.20 instead of 0.13) as predicted by Eq. (12). This discrepancy is thought to occur because, unlike chromatography, where the time scale for absorption of compound to the walls is much shorter than the time scale for flow in the tube, in the tubing experiments the time scales are comparable.

The dependence of tubing delays measured for 2-ketones and 1-alkenes on $c^*$ for a range of conditions are shown in Fig. 4.

Because delays increase proportionally with increasing tubing length, they are plotted as minutes of delay per meter of tubing. Delays are inversely proportional to $c^*$, eventually levelling off at the residence time of the tubing when $c^* \gg C_w$ and gas-wall partitioning becomes insignificant. This trend is driven by the change in the time scale to reach partitioning equilibrium with $c^*$, as described in Eq. (11). Fig. 4 also compares model output to our experimental results, and shows that the model accurately predicts the tubing delay as a function of $c^*$ across all functional groups, tubing lengths, and flow rates

tested.

Model simulations were also conducted for a range of flow rates and $c^*$ typically encountered in laboratory and field studies. The tubing delays predicted by the model are presented in Fig. 5 and are clearly significant, especially for organic compounds with $c^*$ below $10^5\,\mu g\,m^{-3}$. The results also quantify the heuristics already being used by researchers to minimize tubing delays, which are to minimize tubing length, increase flow rate, and heat tubing. Heating tubing increases the $c^*$ of

the compounds being measured, reducing their gas-wall partitioning coefficient and thereby decreasing tubing delay. Sampling compounds through the tubing at higher flows than necessary for instruments and dumping excess flow (oversampling) also reduces tubing delays by decreasing the time available for compounds to partition to the walls. The delays for 3/16 in. ID Teflon tubing presented in Fig. 5 can be estimated using the empirical parameterization in Eq. 13:

$$Delay\ (min\ m^{-1}) = \frac{3.18 \times 10^{-3}}{\frac{Q}{4.73 + Q} + \frac{c^*}{8.11 \times 10^6 + c^*}} \tag{13}$$

where $Q$ is flow rate (L min$^{-1}$) and $c^*$ is the saturation vapor concentration ($\mu g\ m^{-3}$) at the temperature of the tubing calculated using SIMPOL.1. This parameterization matches the model predictions of tubing delay within a factor of 1.2 for delays between 5 sec m$^{-1}$ and 60 min m$^{-1}$ across the range of flow rates and $c^*$ plotted in Fig. 5. We note that although the



parameterization in Eq. 13 is based on diffusion coefficients $D$ estimated at 23 °C, changes in $D$ due to temperature produce only small changes in predicted delay (~10% when raising temperature from 23 °C to 100 °C). This effect is negligible compared to the change in delay caused by the accompanying shift in $c*$ (several orders of magnitude for the same temperature change), making this parameterization a useful predictive tool for the changes in tubing delay caused by changes in temperature. The parameterization does not, however, take into account changes in the absorptive properties of Teflon that may occur at lower or higher temperatures.

### 3.2 Estimating $C_w$ for Teflon tubing

As mentioned above, the value of $C_w$ used in the model for the 0.47 cm ID PFA Teflon tubing was estimated by fitting the model predictions to the experimental data in Fig. 4. The optimal value for $C_w$ was 4 g m$^{-3}$ (grams of absorbing phase per m$^3$ of internal tube volume). In order to directly compare $C_w$ in tubing and chambers one must correct for differences in the surface area to volume ratios. Since we model gas-wall partitioning as occurring within a finite depth at the surface of the Teflon tubing or chamber, the volume of Teflon into which partitioning occurs is the product of the Teflon surface area ($SA$) and the partitioning depth ($\delta$). $C_w$ can then be expressed as in Eq. 14:

$$C_w = \frac{SA\,\delta\,\rho}{V} \tag{14}$$

where $\rho$ is the density of Teflon and $V$ is the volume of gas exposed to the given surface area of Teflon. The equivalent wall mass measured for tubing can then be scaled for comparison with chamber values using Eq. 15:

$$C_{w,ch} = C_{w,t}\frac{SA_{ch}\,V_t\,\delta_{ch}\,\rho_{ch}}{SA_t\,V_{ch}\,\delta_t\,\rho_t} \tag{15}$$

where the subscripts $ch$ and $t$ denote the chamber and tubing. As discussed above, $\delta_{ch}/\delta_t \sim 0.7$–2, and since PFA (tubing) and FEP (chambers) Teflon have the same density, $\rho_{ch}/\rho_t = 1$.

When scaled according to Eq. 15, the value of $C_{w,t} = 4$ g m$^{-3}$ reported above is equivalent to ~10–30 mg m$^{-3}$ of liquid organic aerosol in an 8 m$^3$ chamber. This is comparable to the values of $C_w$ determined in FEP Teflon chambers: 16 mg m$^{-3}$ for 1-alkenes and 24 and 78 mg m$^{-3}$ for 2-ketones (Matsunaga and Ziemann, 2010; Yeh and Ziemann, 2015), where values from Matsunaga and Ziemann (2010) were recalculated with $c*$ values obtained using SIMPOL.1 (Pankow and Asher, 2007). The similarity in $C_w$ values indicates that gas-wall partitioning of organic compounds is similar for PFA and FEP Teflon. We also note that since gas-wall partitioning in tubing depends on $C_w$ (Eq. 6), and since estimates of $C_w$ depend on the method used to estimate compound vapor pressures, researchers applying the results of this work to other compounds should use SIMPOL.1 to estimate $c*$ values, even when measured vapor pressures are available.

### 3.3 Effect of tubing diameter on tubing delays

We find that tubing delays are shortest for small diameter tubing, provided that flow rate or Reynolds number is held constant. When the tubing is being depassivated following a step-function decrease in sample concentration one can treat the



residence time of compound in the walls of the tubing ($\tau_w$) as the limiting step in depassivation. By substituting Eqs. 4, 6 and 14 into Eq. 5 one arrives at an expression for the residence time of a compound in the walls of the tubing given in Eq. 16:

$$\tau_w = \frac{C_w}{k_a \, c^*} = \frac{\delta_t \, \rho_t \, d_t}{2 \, c^* \, D_g} \tag{16}$$

which shows a linear relationship between $\tau_w$ and tubing diameter. While this approach is clearly a simplification and ignores

the effect of flow rate on the rate of equilibration, we show in Fig. S2 that our numerical model also predicts a linear relationship between tubing delay and tubing diameter when flow rate is held constant. This linear relationship also aligns well with our experimental results. In Fig. S3 we present tubing delays and model results for 1/16 and 3/16 in. ID tubing. In the region where delays are dominated by gas-wall partitioning inside the tubing ($c^* < 10^6 \, \mu g \, m^{-3}$) we observe that modeled and measured delays are three times longer for 3/16 in. ID tubing compared to 1/16 in. ID tubing. This relationship breaks

down at higher volatilities ($c^* > 10^6 \, \mu g \, m^{-3}$) since $c^* > C_w$, and the extent of gas-wall partitioning is small, giving very short residence times in the walls. With this in mind, one can conclude that tubing delays scale directly with tubing diameter at a constant flow rate when delays are larger than a few seconds.

To quantify the effect of simultaneous changes in tubing diameter and flow rate we generated model predictions of tubing delay for a compound with $c^* = 10^5 \, \mu g \, m^{-3}$ across a range of tubing diameters and flow rates. These results are presented in

Fig. S4, and provide guidance for designing a sampling system that minimizes tubing delay. As expected, the model predictions indicate that smaller diameter tubing has shorter tubing delays when flow rate or Reynolds number is held constant. The results in Fig. S4 also suggest that one may be able to reduce tubing delays by sampling under turbulent flow conditions. We did not attempt to quantify delays under such conditions, however, in part due to the larger pressure drops experienced in that flow regime, which are undesirable under many circumstances.

**3.4 Effect of instrumentation on delays**

In these experiments gas-wall partitioning inside the PTR-MS was the largest source of delay. We used the PTR-MS time profiles shown in Fig. 2B to quantify the dependence of this delay on c*, and the results are shown with the tubing delays in Fig. 4. The instrument delays observed for the PTR-MS are significant, equivalent to several meters of Teflon tubing. This amount of Teflon is significantly larger than the size of the PTR-MS inlet, indicating that the surfaces inside the PTR-MS are

stickier than Teflon. We encourage others to use this method to determine delays caused by their particular instrument across the range of volatilities being measured. This volatility-dependent response function is important for interpreting the time profiles of compounds being measured, since it sets the lower limit for how fast the instrument can respond to sudden changes in concentration. One can also convolve the instrument response function with the output of the tubing model to quantify the total delays caused by tubing and the instrument, as described in SI.





## 4 Conclusions

We found that gas-wall partitioning of organic compounds inside Teflon tubing significantly affects time-resolved measurements of compounds with saturation concentration ($c*$) below $10^7$ µg m$^{-3}$. The compounds measured in this study (SIMPOL.1-calculated $c*$ values ranging from $3 \times 10^4$ µg m$^{-3}$ to $1 \times 10^7$ µg m$^{-3}$) are all expected to exist entirely in the gas

phase in the atmosphere (Donahue et al, 2006). We found that measurements of compounds in this volatility range were significantly affected by delays caused by partitioning to the walls of the tubing and the PTR-MS. We modeled the delays caused by Teflon tubing using a simple chromatography model and the gas-wall partitioning framework of Matsunaga and Ziemann (2010). This model accurately predicts tubing delays across all compound volatilities, functional groups, tubing lengths and diameters, and flow rates tested. The measurements and model simulations indicate that delays can shift

compound time profiles by minutes to hours – time scales that are highly relevant to both laboratory and atmospheric measurements.

A potential application of these delays is for estimating compound saturation concentration, even when the identity of the compound is unknown. Past work has proposed using desorption kinetics inside an inlet as a technique for identifying compounds (Schuhfried et al., 2012), and the model presented here can be used in a similar way. Another possible

application is to induce time separation among different compounds that are otherwise indistinguishable to the analytical instrumentation (e.g., compounds with the same accurate mass in CIMS; Stark et al., 2015), since the $c*$ of multifunctional compounds with the same molecular formula can often differ by 5 orders-of-magnitude (Krechmer et al., 2015). The separation in time can then be exploited via manual analyses or factor analysis techniques (e.g., Ulbrich et al., 2009). This is equivalent to a "poor-person's chromatography," but using ambient temperature and Teflon surfaces that avoid thermal

decomposition of multifunctional species that can occur in gas chromatography due to use of high temperatures (Stark et al., 2017).

Accounting for tubing and instrument delays is especially important in circumstances where concentrations of the compounds are changing rapidly, including eddy covariance measurements, mobile platforms such as aircraft, rapidly changing emission sources such as fires or motor vehicle exhaust, and fast processes such as chemical reactions and gas-

particle-wall partitioning. As CIMS techniques continue to be developed for detecting multifunctional organic compounds, extra care must be taken to minimize the impact that tubing and instrument surfaces have on measurements since these compounds are especially sticky.

Based on our results, we recommend that studies measuring intermediate volatility and semivolatile compounds ($c* < 10^6$ µg m$^{-3}$) minimize the length of Teflon tubing used; and since increased flow rates and smaller tubing diameters also help to

mitigate tubing delays, use the fastest flow rate in the smallest diameter tubing that still maintains laminar flow. We also recommend that researchers determine the volatility-dependent time response function of their instrument. This sets the lower limit for the response time for a given compound and allows for deconvolution of the effects of gas-wall partitioning in the instrument from those in the inlet tubing. The instrument response function can then be convolved with the output of



the model presented in this study to correct for delay artifacts caused by gas-wall partitioning in Teflon tubing. This method can also be used to characterize other tubing materials, the effect of temperature and other variables, and enable improved inlet system designs.

## Code & data availability

Data for each figure and model code will be available for download (after the paper is accepted) at: http://cires1.colorado.edu/jimenez/group_pubs.html

## Competing interest

The authors declare that they have no conflict of interest

## Acknowledgements

We thank the Alfred P. Sloan Foundation (Grant No. G–2013–6–02), the US DOE (BER/ASR, Grant DE-SC0016559), and the National Science Foundation (Grants AGS-1420007 and AGS-1360834) for funding this study. One of us (JdG) was a consultant for Aerodyne Research Inc. during part of the study.

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



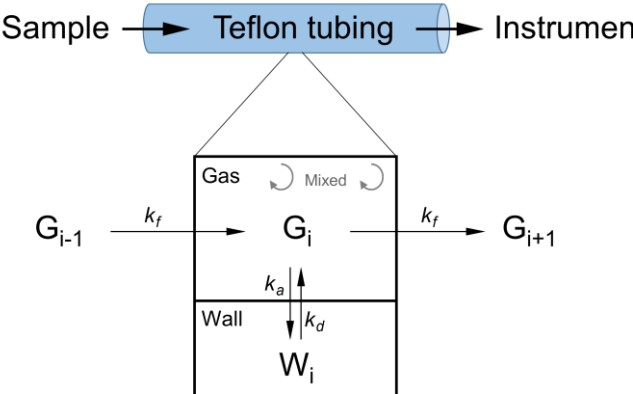

**Figure 1.** Schematic of the chromatography model used to describe delays caused by gas-wall partitioning of organic compounds in Teflon tubing. Compounds flow through a series of perfectly mixed bins, undergoing gas-wall partitioning within each bin. The rates of flow between bins ($k_f$), absorption ($k_a$) and desorption ($k_d$) are dependent on tubing diameter, flow rate, and the saturation vapor concentration $c*$ of the compound being measured.



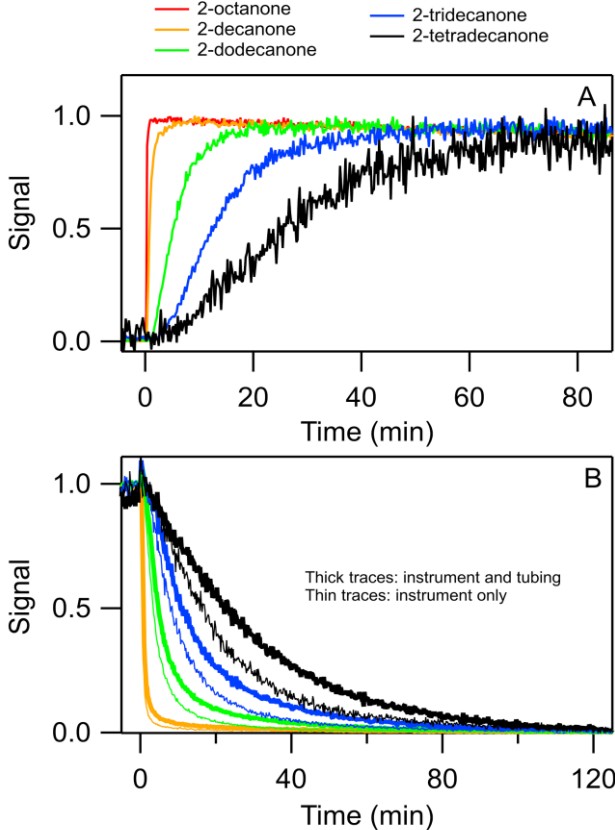

**Figure 2.** (a) PTR-MS time profiles measured in response to a step function increase in the concentration of 2-ketones. All compounds were measured simultaneously. Profiles are normalized to peak signal. (b) PTR-MS time profiles measured in response to a step function decrease in the concentration of 2-ketones for tubing + PTR-MS (thick lines) and the PTR-MS alone (thin lines). Profiles are normalized to the equilibrium concentration measured prior to the step change.





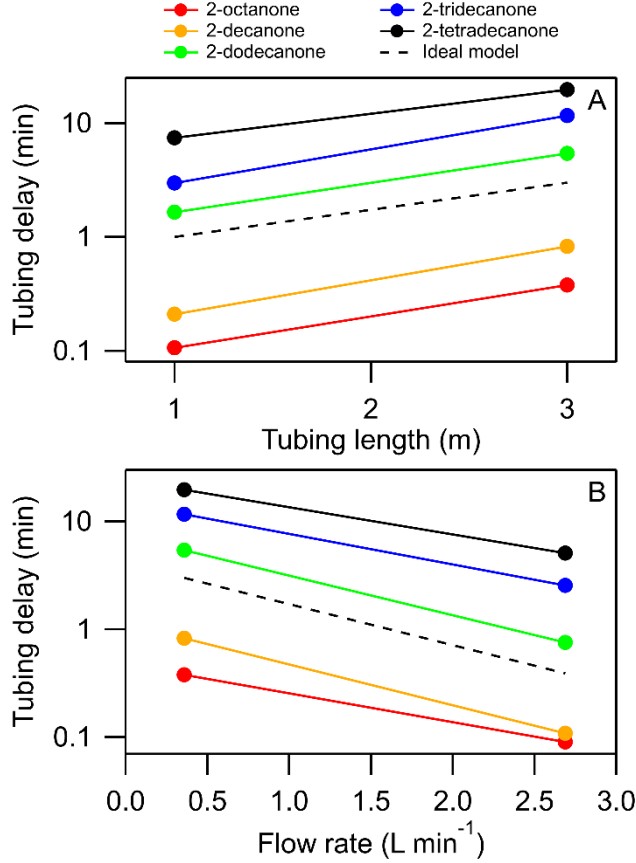

**Figure 3.** (a) Tubing delays for PTR-MS analysis of 2-ketones sampled using 1 and 3 m of 3/16 in. ID tubing at a flow rate of 0.36 L min$^{-1}$. The dashed line corresponds to the model case where the delay is proportional to the length of the tubing, which for these experiments is a factor of three. (b) Tubing delays for PTR-MS analysis of 2-ketones sampled using 3 m of 3/16 in. ID tubing and flow rates of 0.36 and 2.7 L min$^{-1}$. The dashed line corresponds to the model case where the delay is inversely proportional to the flow rate, which for these experiments is a factor of 0.13.





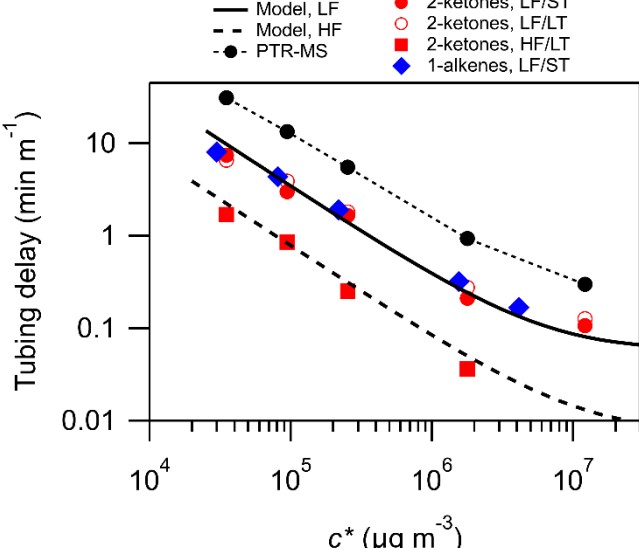

**Figure 4.** Comparison of modeled and measured tubing delays for homologous series of 2-ketones and 1-alkenes with a range of $c*$ values using short (ST = 1 m) and long (LT = 3 m) lengths of 3/16 in. ID tubing with low (LF = 0.36 L min$^{-1}$) flow, and long tubing with low and high (HF = 2.7 L min$^{-1}$) flow. The instrument delay for the PTR-MS is also shown. Values of $c*$ were calculated using SIMPOL.1 (Pankow and Asher, 2007).



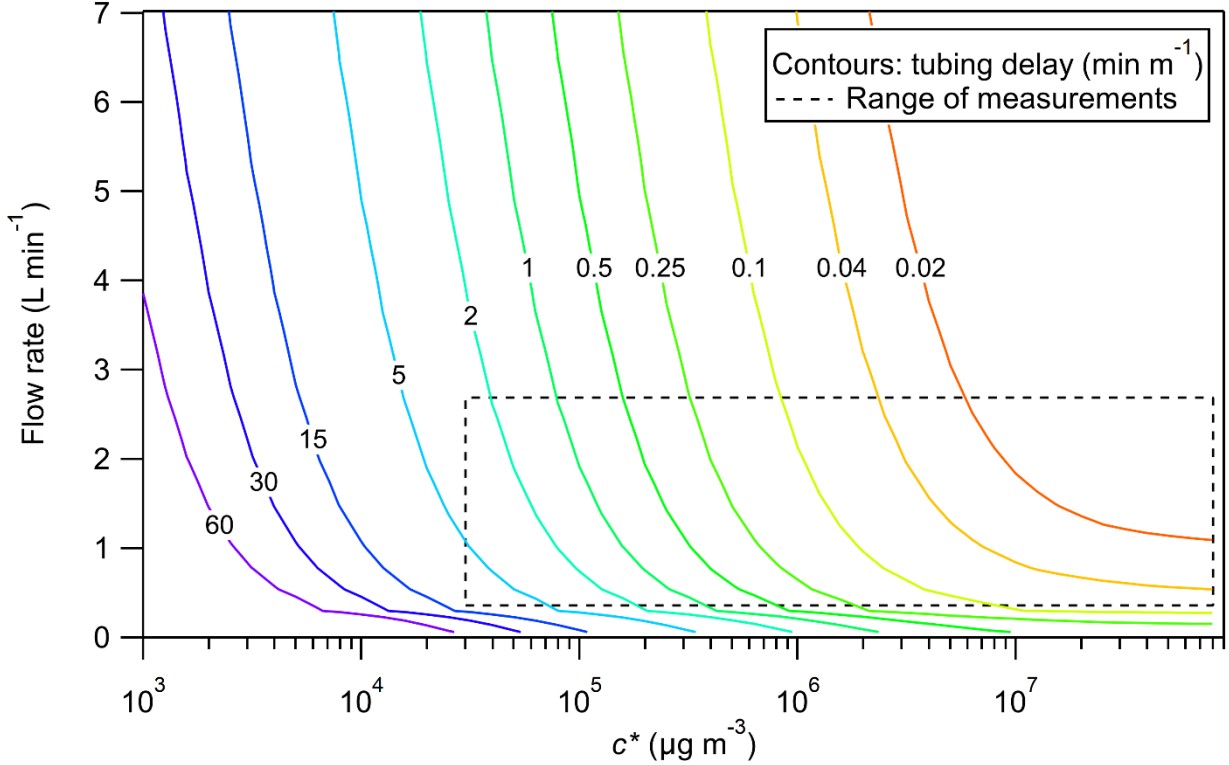

**Figure 5.** Simulated tubing delays presented as a function of $c*$ and flow rate for 3/16 in. ID PFA Teflon tubing. The range of conditions for measurements made in this study are shown by the dashed box. Values of $c*$ were calculated using SIMPOL.1 (Pankow and Asher, 2007).