# Peer review of "Effects of Gas-Wall Partitioning in Teflon Tubing and Instrumentation on Time-Resolved Measurements of Gas-Phase Organic Compounds"

_Atmospheric Measurement Techniques, 2017_

## Referee Comment (RC1) · Anonymous Referee #1 · 18 Sep 2017

This paper presents a combined laboratory and model study on the tubing delay of homologous series of long-chain alkenes and ketones in Teflon tubing as well as in a PTR-MS instrument. A model based on a linear chromatography model was developed, which generally compared well with results of gas-wall partitioning experiments. The authors found that delays vary proportionally with tubing diameter and length, and inversely with saturation concentration and flow rate. The delay caused by the PTR-MS itself without tubing was relatively large, showing that surfaces inside the instrument behave "stickier" than Teflon. The paper raises an interesting topic for the community, because Teflon tubing is widely used in laboratories and field studies, and PTR-MS instruments are often used for fast-response measurements. The authors give indica-

tions on how to improve sampling and recommend users to characterize tubing and instrument delays.

The paper is well written using a clear language, and is suitable for publishing in AMT after the authors have addressed the following comments:

1) The experiments were exclusively conducted for long-chain alkenes and ketones. What was the reason to choose these classes of compounds? They do not seem to be typical atmospheric semivolatile compounds generally measured by PTR-MS.

2) p.3, l.6: The authors use 20 ppb mixing ratio for all experiments. What was the reason to choose this mixing ratio? What would be the influence of using other mixing ratios?

3) p. 5, l. 15: A linear fitting of the largest dataset obtained in the study for deriving an optimum chamber wall mass concentration is mentioned here. Some clarification would help to understand what was done: Which quantities were fitted against each other? How good was the correlation, and how exactly was the optimum value derived?

4) p. 6, l. 16: Were the experiments conducted just once? How is the reproducibility of the experiments?

5) p. 10, l. 25: It is mentioned here that CIMS techniques are used for detecting multifunctional groups and that these provide challenges due to their stickiness. The experiments conducted for this paper focused on alkenes and ketones, which are not multifunctional. Would the authors expect their approach to be suitable as well for compounds containing other functional groups, such as carboxylic acids or aldehydes?

6) Fig. 3: The plots look like linear fits through just two data points. Do you expect the relationship to be linear for more data points? Is there any experimental data to confirm this?

7) In the title, the authors use the general word "instrumentation". Maybe it would be more precise to replace this with "PTR-MS", as this is the instrumentation for which the

study was conducted.

---

## Referee Comment (RC2) · Anonymous Referee #2 · 27 Sep 2017

This manuscript describes the gas-wall partitioning of organic compounds in Teflon tubing and "real-time" instruments using Teflon materials such as PTR-MS. Rapid partitioning of compounds with $c^*$ in the range of 10e4 to 10e7 micro g /m3 causing delays in instrument response. These delays were measured for 2-ketones and 1-alkenes falling into the respective $c^*$ range as a function of tubing length and flow rate. The delays were also modelled using a simple chromatography model and the gas-wall partitioning framework of Matsunaga and Ziemann (2010). The model predicts tubing delays in agreement with experimental results concluding that compound time profiles can be shifted by minutes to hours. These time delays have been observed and described already earlier. See and include the following reference: Schaub et al. 2010

(doi:10.1111/j.1399-3054.2009.01322.x) Response times of sesquiterpenes and green leaf volatiles were investigated in a plant cuvette + Teflon tubing at different temperatures.

In the present manuscript additional effort was undertaken to describe observed time delays with a model. This is very useful for the atmospheric measurement community drying to measure not only volatile- but also semi- and non-volatile organics in the gas phase.

The manuscript is well written and might be suitable for publishing in AMT after the authors have addressed the following comments:

All experiments were performed under dry conditions and I wonder how relative humidity (RH) will impact the delay time of different compounds. I would like to see either RH results included in the manuscript or a paragraph discussing this issue. Measurements in the real atmosphere and also in smog chambers contain a substantial amount of RH and additional solvation effects of water adsorbed into the Teflon walls might affect organic compounds differently. Discuss in detail if and how your "dry" results can be applied to real world measurements.

Page 6, lines 15-20: You describe time profiles shown in Fig. 2a, signal (y-axis) normalized product ion count rates as a function of time (x-axis). In the sentence before you talk about "delays" that were quantified in this study ....90% of the total change... confusing, please reorganize. Fig. 2a: Explain why the signals of 2-octanone, 2-decanone, and 2-dodecanone steadily decline after the sharp step function increase. Explain the "noise" especially for 2-tetradecanone. Discuss reproducibility of individual measurements. Present error estimates for individual compounds. Fig. 2b: 2-octanone is missing

Mention details such as line dimension and flow rate in the figure caption.

Page 7, lines 5-20: The flow velocity has a radial profile in a 0.47 cm ID tube. Which

flow velocity is used and why?

In the text, caption and figure 2,3, use either "a" or "A"( "b" or "B")
* * *

---

## Author Comment (AC1) · 13 Oct 2017

Response to referee comments for:

**Effects of Gas-Wall Partitioning in Teflon Tubing and Instrumentation on Time-Resolved Measurements of Gas-Phase Organic Compounds**

Demetrios Pagonis[1,2], Jordan E. Krechmer[1,2,3], Joost de Gouw[1,4], Jose L. Jimenez[1,2], and Paul J. Ziemann[1,2]

[1]Cooperative Institute for Research in Environmental Sciences (CIRES), Boulder, Colorado 80309, United States

[2]Department of Chemistry and Biochemistry, University of Colorado, Boulder, Colorado 80309, United States

[3]Aerodyne Research, Inc., Billerica, Massachusetts 01821, United States

[4]NOAA Earth System Research Laboratory, Boulder, Colorado 80305, United States

We thank both referees for their reviews - our responses to each comment are below. Referee comments are in bold face.

**Referee #1**

**R1.1) The experiments were exclusively conducted for long-chain alkenes and ketones. What was the reason to choose these classes of compounds? They do not seem to be typical atmospheric semivolatile compounds generally measured by PTR-MS.**

We chose homologous series of 1-alkenes and 2-ketones for this study since standards are readily available for a wide range of carbon numbers, and because their gas-wall partitioning in Teflon chambers has already been studied, allowing us to compare the partitioning in PFA Teflon tubing to that in FEP Teflon atmospheric chambers. Our results show that the extent of gas-wall partitioning in Teflon tubing is largely determined by compound volatility, so we believe that our results are widely applicable to semivolatile organic compounds in the atmosphere.

We have updated page 7, line 20 to include the following sentence:

*"No effect of functional group on tubing delay was observed, consistent with past studies of gas-wall partitioning in Teflon chambers, where compound volatility is the property that best predicts the extent of sorption (Matsunaga and Ziemann 2010)."*

**R1.2) p.3, l.6: The authors use 20 ppb mixing ratio for all experiments. What was the reason to choose this mixing ratio? What would be the influence of using other mixing ratios?**

We chose a mixing ratio of 20 ppb to balance sufficient signal-to-noise ratio with the need to not deplete the reagent ion of the PTR-MS. Past work has shown that the equilibrium constant for gas-wall partitioning in Teflon chambers remains constant across a wide range of mixing ratios, indicating that the total VOC load does not affect partitioning to Teflon (Matsunaga and Ziemann, 2010).

We have added the following discussion to page7, line 21:

*"We note here that past work has shown that gas-wall partitioning equilibrium established in Teflon chambers is independent of sample concentration (Matsunaga and Ziemann, 2010) and relative humidity (Krechmer et al., 2017). Limited experiments conducted here were consistent with those findings, indicating that this model can be used to estimate tubing delays in both lab and field settings."*

**R1.3) p. 5, l. 15: A linear fitting of the largest dataset obtained in the study for deriving an optimum chamber wall mass concentration is mentioned here. Some clarification would help to understand what was done: Which quantities were fitted against each other? How good was the correlation, and how exactly was the optimum value derived?**

We agree that more discussion of the fitting procedure is useful. We have added the following section to the SI, page 3:

*"**Fitting procedure to estimate $C_w$**

The effective wall mass $C_w$ for PFA Teflon tubing was determined by fitting the model output for our base experimental case (0.47 cm ID tubing sampling at 0.36 L min$^{-1}$) to the experimentally determined tubing delays under those conditions. We varied $C_w$ in the model to generate tubing delays as a function of c\*, and we then used an orthogonal-distance regression to minimize a sum-of-squares residual. These residuals were calculated in log-log space (log(delay) vs log(c\*), as the data is shown in Fig. 4) because both compound c\* and tubing delays vary across several orders of magnitude and we wished to avoid biasing the fitting towards data acquired at high c\* or at longer tubing delays. The $C_w$ value assumed by the model was then varied to find the minimum residual, giving an optimal $C_w$ value of 4 g m$^{-3}$. The delays predicted for this $C_w$ value are plotted in Fig. 4."*

We have also added the following sentence to page 5, line 16:

*"Additional details on the fitting procedure used to determine $C_w$ can be found in the SI of this paper."*

**R1.4) p. 6, l. 16: Were the experiments conducted just once? How is the reproducibility of the experiments?**

We estimate from our replicates that delays vary by 10% for long delays, and due to the sampling rate of the PTR-MS we cannot achieve uncertainties below 5 seconds for short delays. We agree with both reviewers that the reproducibility was not well addressed in the manuscript. We have added error bars to Fig. 4 for the data set that has the most replicates (1 m of PFA tubing sampling 2-ketones at 0.37 L min$^{-1}$):

[Figure]

*"**Figure 4.** Comparison of modeled and measured tubing delays for homologous series of 2-ketones and 1-alkenes with a range of c\* values using short (ST = 1 m) and long (LT = 3 m) lengths of 3/16 in. ID tubing with low (LF = 0.36 L min$^{-1}$) flow, and long tubing with low and high (HF = 2.7 L min$^{-1}$) flow. The instrument delay for the PTR-MS is also shown. Values of c\* were calculated using SIMPOL.1 (Pankow and Asher, 2007). Error bars are only shown for the 2-ketones LF/ST case, and represent the variability (std. dev.) of the observations. Note that for the larger delays, the error bars are smaller than the data points."*

**R1.5) p. 10, l. 25: It is mentioned here that CIMS techniques are used for detecting multifunctional groups and that these provide challenges due to their stickiness. The experiments conducted for this paper focused on alkenes and ketones, which are not multifunctional. Would the authors expect their approach to be suitable as well for compounds containing other functional groups, such as carboxylic acids or aldehydes?**

Past work measuring the equilibrium constants of organic compounds partitioning into Teflon films has shown that the extent of partitioning is determined by compound volatility (see summary in Fig. 4 of Krechmer et al., 2016). Those measurements included carboxylic acids and diols that were added to chambers as standards, as well as multifunctional hydroxynitrates that were produced *in-situ*. In this study we saw that the gas-wall partitioning of ketones and alkenes was the same in Teflon tubing as it is in Teflon chambers, so we expect that carboxylic acids and other multifunctional compounds would show delays consistent with those predicted by our model. We address this in conjunction with comment R1.1 on page 7.

**R1.6) Fig. 3: The plots look like linear fits through just two data points. Do you expect the relationship to be linear for more data points? Is there any experimental data to confirm this?**

Yes, the lines plotted in Fig. 3 directly connect the two data points for each compound. In Fig. 3A we expect the relationship to be linear for additional tubing lengths, and in Fig. 3B we expect measurements at additional flow rates to show an inverse relationship. While our data is limited to the measurements of five compounds across two flow rates and two tube lengths, there is extensive data in the chromatography literature supporting our conclusions, which are summarized by Eq. 12 in the manuscript (Skoog, et al. 2017). We have modified the sentence on page 7, line 5 to read:

*The tubing delay increases almost proportionally with tubing length, similar to the effect of column length on retention time in chromatography captured in Eq. (12):... where $t_r$ is retention time, L is column length, $v_f$ is flow velocity, and B is a constant that incorporates the partitioning coefficient and volumes of stationary and mobile phases (Poole, 2003; Skoog et al., 2007).*

**R1.7) In the title, the authors use the general word "instrumentation". Maybe it would be more precise to replace this with "PTR-MS", as this is the instrumentation for which the study was conducted.**

The choice of "instrumentation" in the title was deliberate as the approach we outline in the manuscript for characterizing instrument response times can be widely applied to any instrument making time-resolved measurement of gas-phase organic compounds. Accordingly, we have opted to keep the more general "instrumentation" in the title.

**Referee #2**

**R2.1) All experiments were performed under dry conditions and I wonder how relative humidity (RH) will impact the delay time of different compounds. I would like to see either RH results included in the manuscript or a paragraph discussing this issue. Measurements in the real atmosphere and also in smog chambers contain a substantial amount of RH and additional solvation effects of water adsorbed into the Teflon walls might affect organic compounds differently. Discuss in detail if and how your "dry" results can be applied to real world measurements.**

It has been shown that relative humidity does not affect the gas-wall partitioning equilibrium in Teflon chambers (Krechmer et al. 2017, SI), and limited tests conducted in our lab showed no effect of relative humidity on depassivation times for Teflon tubing. Accordingly, we expect that our results are widely applicable to measurements in humid environments.

We have added the following discussion to page 7, line 21:

*"We note here that past work has shown that gas-wall partitioning equilibrium established in Teflon chambers is independent of sample concentration (Matsunaga and Ziemann, 2010) and relative humidity (Krechmer et al., 2017). Limited experiments conducted here were consistent with those findings, indicating that this model can be used to estimate tubing delays in both lab and field settings."*

**R2.2) Page 6, lines 15-20: You describe time profiles shown in Fig. 2a, signal (y-axis) normalized product ion count rates as a function of time (x-axis). In the sentence before you talk about "delays" that were quantified in this study . . ..90% of the total change. . . confusing, please reorganize.**

We have clarified this section by directing the reader to Fig. 2 prior to discussing how delays were quantified. The sentence beginning on page 6, line 15 now reads:

*"Tubing delays were measured by introducing step function changes in the concentration of organic compounds measured by the PTR-MS, with all compounds of a homologous series being measured simultaneously, as shown in Fig. 2."*

**R2.3) Fig. 2a: Explain why the signals of 2-octanone, 2- decanone, and 2-dodecanone steadily decline after the sharp step function increase. Explain the "noise" especially for 2-tetradecanone. Discuss reproducibility of individual measurements. Present error estimates for individual compounds.**

The decline in signal over time observed here is due to drift in instrument response following start-up. Instrument response typically stabilized following the initial passivation of the sampling line and instrument surfaces (shown in Fig. 2a).

The decrease in signal-to-noise ratio observed for higher carbon numbers is attributable to the lower concentration of analyte present in the chamber as well as the mass discrimination of the quadrupole mass analyzer in the PTR-MS. Equimolar amounts of each ketone were injected into the chamber for each experiment, and the larger ketones undergo gas-wall partitioning inside the Teflon chamber. Past work indicates that ~50% of the 2-tetradecanone is expected to be in the chamber walls at equilibrium (Yeh and Ziemann, 2015). All other factors being equal, we would expect 2-tetradecanone to have the lowest signal-to-noise ratio as a result of it being present in the lowest concentration in the chamber. The lower signal-to-noise ratios for the larger ketones are also due to the mass transmission efficiency of the quadrupole mass analyzer used in the PTR-MS. The mass analyzer used in this study is engineered to transmit ions with $m/z$ up to 250. Protonated 2-tetradecanone is detected at $m/z$ 213, so we expect that mass discrimination in the quadrupole mass analyzer will decrease the signal-to-noise ratio for the larger ketones studied here. We have modified the caption of Fig. 2 to read as follows:

*Figure 2. (A) PTR-MS time profiles measured in response to a step function increase in the concentration of 2-ketones. All compounds were measured simultaneously through 1 m of 3/16 in. ID PFA Teflon tubing at a flow rate of 0.36 L min$^{-1}$. Profiles are normalized to peak signal. The decline in signal over time is due to drift in instrument response following start-up.*

*(B) PTR-MS time profiles measured in response to a step function decrease in the concentration of 2-ketones for tubing + PTR-MS (thick lines) and the PTR-MS alone (thin lines). The tubing used was a 3 m length of 3/16 in. ID PFA Teflon, and the flow rate for both traces was 0.36 L min $^{-1}$. Profiles are normalized to the equilibrium concentration measured prior to the step change. The signal-to-noise ratio is lower at higher carbon numbers due to gas-wall partitioning in the chamber lowering sample concentration as well as mass discrimination within the PTR-MS quadrupole mass analyzer. For visual clarity the traces for 2-octanone are not shown since they overlap with the traces for 2-decanone.*

We have added error estimates to Fig. 4. Additional discussion and the new version of Fig. 4 are above, alongside comment R1.4 from Referee 1.

**R2.4) Fig. 2b: 2-octanone is missing Mention details such as line dimension and flow rate in the figure caption.**

We chose to omit 2-octanone for visual clarity as the traces overlaps significantly with the traces for 2-decanone. We have updated the figure caption for Fig. 2 to include line dimensions, flow rates, and discussion of why the traces for 2-octanone are omitted in panel B. The updated caption is above, alongside comment R2.3.

**R2.5) Page 7, lines 5-20: The flow velocity has a radial profile in a 0.47 cm ID tube. Which flow velocity is used and why?**

The flow velocity used in this discussion is the bulk flow velocity. In our model we do not account for the radial profile of flow velocities under laminar flow conditions. This allows us to model the tubing as a series of perfectly-mixed bins, which greatly simplifies the computations. We discuss the errors introduced by this assumption in the SI of the paper.

We have updated page 7, line 8 to clarify that we are discussing bulk flow velocity.

**R2.6) In the text, caption and figure 2,3, use either "a" or "A" ( "b" or "B")**

All instances updated to capital A/B.

**References**

Krechmer, J. E., Pagonis, D., Ziemann, P. J. and Jimenez, J. L.: Quantification of Gas-Wall Partitioning in Teflon Environmental Chambers Using Rapid Bursts of Low-Volatility Oxidized Species Generated in Situ, Environ. Sci. Technol., 50, 5757–5765, doi:10.1021/acs.est.6b00606, 2016.

Krechmer, J. E., Day, D. A., Ziemann, P. J. and Jimenez, J. L.: Direct Measurements of Gas/Particle Partitioning and Mass Accommodation Coefficients in Environmental Chambers, Environ. Sci. Technol., acs.est.7b02144, doi:10.1021/acs.est.7b02144, 2017.

Matsunaga, A. and Ziemann, P. J.: Gas-Wall Partitioning of Organic Compounds in a Teflon Film Chamber and Potential Effects on Reaction Product and Aerosol Yield Measurements, Aerosol Sci. Technol., 44, 881–892, doi:10.1080/02786826.2010.501044, 2010.

Poole, C. F.: The essence of chromatography, 1st ed., Elsevier, Boston, MA., 2003.

Skoog, D. A., Holler, F. J. and Crouch, S. R.: Principles of Instrumental Analysis, Cengage Learning, Boston, MA., 2006.

Yeh, G. K. and Ziemann, P. J.: Gas-Wall Partitioning of Oxygenated Organic Compounds: Measurements, Structure–Activity Relationships, and Correlation with Gas Chromatographic Retention Factor, Aerosol Sci. Technol., 49, 727–738, doi:10.1080/02786826.2015.1068427, 2015.